# Gut Microbiome Associated with the Psychoneurological Symptom Cluster in Patients with Head and Neck Cancers

**DOI:** 10.3390/cancers12092531

**Published:** 2020-09-06

**Authors:** Jinbing Bai, Deborah Watkins Bruner, Veronika Fedirko, Jonathan J. Beitler, Chao Zhou, Jianlei Gu, Hongyu Zhao, I-Hsin Lin, Cynthia E. Chico, Kristin A. Higgins, Dong M. Shin, Nabil F. Saba, Andrew H. Miller, Canhua Xiao

**Affiliations:** 1Nell Hodgson Woodruff School of Nursing, Emory University, Atlanta, GA 30322, USA; deborah.w.bruner@emory.edu; 2Rollins School of Public Health, Winship Cancer Institute, Emory University, Atlanta, GA 30322, USA; veronika.fedirko@emory.edu; 3Department of Radiation Oncology, School of Medicine, Emory University, Atlanta, GA 30322, USA; jjbeitl@emory.edu (J.J.B.); kristin.higgins@emory.edu (K.A.H.); 4Department Biostatistics, Yale School of Public Health, Yale University, New Haven, CT 06520, USA; zhou.chao@yale.edu (C.Z.); jianlei.gu@yale.edu (J.G.); hongyu.zhao@yale.edu (H.Z.); 5Department of Epidemiology and Biostatistics, Memorial Sloan-Kettering Cancer Center, New York, NY 10017, USA; LinI1@mskcc.org; 6Department of Psychiatry and Behavioral Sciences, School of Medicine, Emory University, Atlanta, GA 30322, USA; cynthia.chico@emory.edu (C.E.C.); amill02@emory.edu (A.H.M.); 7Department of Hematology and Medical Oncology, School of Medicine, Emory University, Atlanta, GA 30322, USA; dmshin@emory.edu (D.M.S.); nfsaba@emory.edu (N.F.S.); 8School of Nursing, Yale University, New Haven, CT 06477, USA; canhua.xiao@yale.edu

**Keywords:** gut microbiome, head and neck cancer, psychoneurological symptoms, symptom cluster, radiation therapy

## Abstract

**Simple Summary:**

Patients with head and neck cancers (HNCs) report a cluster of psychoneurological symptoms (i.e., the PNS cluster), including pain, fatigue, sleep disturbance, depressive symptoms, and cognitive dysfunction across different treatments. The aim of this longitudinal study was to examine associations between the gut microbiome and the PNS cluster in 13 patients with HNCs pre- and post-radiotherapy. Patients with the high PNS cluster showed a greater decrease in microbial evenness than those with the low PNS cluster from pre- to post-radiotherapy. The high and low PNS clusters showed significant gut microbiome dissimilarities. Patients with the high PNS cluster had higher abundances of *Bacteroidetes, Ruminiclostridium9, Tyzzerella, Eubacterium_fissicatena,* and *DTU089*. Patients with the low PNS cluster had higher abundances in *Lactococcus, Phascolarctobacterium*, and *Desulfovibrio*. Glycan metabolism and vitamin metabolism were different between the high and low PNS clusters pre- and post-radiotherapy.

**Abstract:**

Cancer patients experience a cluster of co-occurring psychoneurological symptoms (PNS) related to cancer treatments. The gut microbiome may affect severity of the PNS via neural, immune, and endocrine signaling pathways. However, the link between the gut microbiome and PNS has not been well investigated in cancer patients, including those with head and neck cancers (HNCs). This pilot study enrolled 13 patients with HNCs, who reported PNS using the Patient-Reported Outcomes version of the Common Terminology Criteria for Adverse Events (CTCAEs). Stool specimens were collected to analyze patients’ gut microbiome. All data were collected pre- and post-radiation therapy (RT). Associations between the bacterial abundances and the PNS clusters were analyzed using the linear discriminant analysis effect size; functional pathway analyses of 16S rRNA V3-V4 bacterial communities were conducted using Tax4fun. The high PNS cluster had a greater decrease in microbial evenness than the low PNS cluster from pre- to post-RT. The high and low PNS clusters showed significant differences using weighted UniFrac distance. Those individuals with the high PNS cluster were more likely to have higher abundances in phylum *Bacteroidetes*, order *Bacteroidales*, class *Bacteroidia*, and four genera (*Ruminiclostridium9, Tyzzerella, Eubacterium_fissicatena*, and *DTU089*), while the low PNS cluster had higher abundances in family *Acidaminococcaceae* and three genera (*Lactococcus, Phascolarctobacterium*, and *Desulfovibrio*). Both glycan metabolism (Lipopolysaccharide biosynthesis) and vitamin metabolism (folate biosynthesis and lipoic acid metabolism) were significantly different between the high and low PNS clusters pre- and post-RT. Our preliminary data suggest that the diversity and abundance of the gut microbiome play a potential role in developing PNS among cancer patients.

## 1. Introduction

Head and neck cancers (HNCs) account for approximately 4% of all cancer diagnosed in the United States [1]. The primary treatments for HNCs include surgery, radiation therapy (RT), and chemotherapy, which is often combined with RT [2]. The purpose of RT is to decrease tumor size to allow surgical resection of tumors, or to kill tumor cells that cannot be removed via surgeries [3]. Patients with HNCs have a high survival rate after RT, especially RT following surgeries [2]. Nevertheless, patients report an array of co-occurring symptoms across different treatments, including pain, fatigue, sleep disturbance, depressive symptoms, and cognitive dysfunction [4,5]. These co-occurring symptoms are defined as the psychoneurological symptoms (PNS) cluster, which potentially have common underlying biological mechanisms [6]. Poor treatment management of the PNS cluster could lead to adverse effects on a patient’s functional performance and may decrease their overall quality of life (QOL) [6].

Understanding the biological mechanism(s) of the PNS cluster can help advance both the diagnosis and treatment of these co-occurring symptoms. Literature has suggested several potential biological mechanisms, including pro-inflammatory response as indicated by higher pro-inflammatory cytokines (e.g., IL-1, IL-6, IL-8, and TNF-α) and the dysregulation of monoamine neurotransmission system (e.g., vagal nerve system) [7]. Additionally, lower estrogen or hemoglobin levels may enhance the development of the PNS cluster [7]. Both pro-inflammatory cytokines and the neurotransmission system may affect, and be modulated by, the critical gut microbiome ecosystem required for psychoneurological health [8,9]. The human gut hosts tens of trillions of microbes, representing 500 species on average [10,11]. The gut microbiome is the collection of all genomes of the microbes in the human gastrointestinal tract [8]. According to the microbiome–gut–brain axis model [12], the gut microbiome may regulate the development of the PNS cluster through neural, immune, metabolic, and hormonal pathways [13,14]. For patients with HNCs, cancer treatments such as RT and chemotherapy can affect patients’ immune responses, reshape the composition of the gut microbiome, and lead to lower abundance of commensal microbes and increased presence of pathogens [15,16], in turn impacting the PNS cluster. The current state of the science regarding the gut microbiome is still at its infancy, and it has not been well investigated in patients treated for cancer, including HNCs.

To date, studies investigating the association between the gut microbiome and PNS have primarily been conducted on animal models and among non-cancer populations. For instance, germ-free (GF) mice exposed to mild restraint stress had an exaggerated release of adrenocorticotropic hormone and corticosterone compared to mice with a healthy microbiome community and no specific pathogens [17]; these results suggest that the gut microbiome has a connection with the brain, and it can cause reduced responses in anxiety and cognitive functions for these mice [17,18,19]. Another study examined whether the absence of a healthy microbiome can influence inflammatory pain and found that GF mice had lower inflammatory hypernociception [20]. Graeau et al. further studied the effects of acute enteric infection or absence of the gut microbiome on anxiety and memory formation in mice, and their work suggested that the gut microbiome influences the ability to form memory, especially for the GF mice [19]. Emerging evidence from these animal studies seems to corroborate the beneficial effects of probiotic or prebiotic treatments as they relate to the PNS cluster [21,22]. Although the preclinical and non-cancer studies provide evidence of an association between the gut microbiome and the PNS cluster, this relationship must be tested in patients with cancer before further interventions can be designed to prevent or alleviate PNS in these patients.

Therefore, the purpose of this study was to examine the associations between the diversity, composition, and function of the gut microbiome and the PNS cluster in patients with HNCs pre- and post-RT. Findings of this study could contribute to understanding the gut microbiome compositional and functional mechanisms underlying the PNS cluster and could provide preliminary insights into designing larger studies to manage treatment symptoms.

## 2. Results

### 2.1. Study Participants Baseline Characteristics

We enrolled 13 patients with an average age of 60 years. Most of them were men (84.6%), White (84.6%), and married (61.5%) (Table 1). This sample represented the HNCs population according to previous literature [23,24]. About half of the participants were diagnosed with oropharyngeal cancer, and 64% of them were HPV-related HNCs. More than half of enrolled patients received chemoradiation and 62% had surgery. 

### 2.2. Taxonomic Profile of the Gut Microbiome and PNS

In total, 26 gut microbial samples from these 13 patients (pre- and post-RT) were analyzed. A median of 137,054 (ranging from 95,849 to 548,061) high-quality sequences per sample were retained after quality control using Divisive Amplicon Denoising Algorithm (DADA2). Rarefaction was performed at 7500 reads based on the frequency per sample. Taxonomic analysis was assigned using a pre-trained Silva 132 99% database. Figure 1 displays the microbial taxa at the phylum level (Figure 1a) and the genus level (Figure 1b) stratified by timepoint (pre- vs. post-radiation therapy) vs. the level of the PNS cluster (low vs. high). The gut microbiome profile in our participants was dominated by *Firmicutes* (43.11%), *Bacteroidetes* (43.83%), and *Actinobacteria* (3.11%) at the phylum level, as well as *Bacteroides* (26.65%), *Parabacteroides* (7.85%), *Blautia* (6.18%), *Alistipes* (6.0%), and *Akkermansia* (5.80%) at the genus level. 

### 2.3. Diversity Analysis of the Gut Microbiome and PNS

The *α*-diversity estimates (i.e., Chao1, Faith’s phylogenetic diversity (Faith PD), Shannon, and evenness index) had no associations with demographic variables (i.e., age in year, gender (male vs. female), race (White vs. non-White), marital status (married vs. unmarried), and history of smoking) or clinical variables (BMI, antibiotics usage (yes vs. no), primary cancer site, cancer stage, cancer treatment, or HPV status (positive vs. negative)). Patients who consumed one or more alcoholic drinks per week (H = 4.938, *p* = 0.026) and indicated no use of anti-inflammatory medications (H = 4.174, *p* = 0.041) had a higher Shannon’s index; Further, patients with a history of one or more alcoholic drinks per week had a higher evenness in their gut microbiomes (H = 5.707, *p* = 0.017). The *β*-diversity estimates showed significant differences in cancer stage for Bray–Curtis distance (F = 1.856, *p* = 0.004, R^2^ = 0.072) and weighted UniFrac distance (F = 2.100, *p* = 0.039, R^2^ = 0.080). No differences in *β*-diversity estimates were found for antibiotic use.

All the *α*-diversity estimates had no associations with the categorical PNS cluster (low vs. high) cross-sectionally (pre- or post-RT). Table 2 shows the longitudinal associations between *α*-diversity estimates (Faith_PD and evenness index) and the categorical PNS cluster. Evenness of the gut microbiome showed a significant association with the PNS cluster over time, with the high level of PNS cluster showing a significantly-larger decrease in microbial evenness than the low level of PNS cluster (*p* = 0.026). The PNS cluster showed a significant interaction with timepoint (pre- and post-RT) for Faith_PD (*p* < 0.0001), showing that relative to pre-RT, the high PNS cluster was associated with lower Faith_PD diversity post-RT.

ADONIS was implemented to explore associations between the *β*-diversity estimates (Bray–Curtis distance and weighted UniFrac distance) and the PNS cluster cross-sectionally. The PNS cluster (low vs. high) showed no significant difference based on the Bray–Curtis distance (Figure 2(a1), F = 1.174, *p* = 0.223, R^2^ = 0.047), but significant differences in the PNS cluster (low vs. high) were found based on the weighted UniFrac distance (Figure 2(b1), F = 2.117, *p* = 0.034, R^2^ = 0.081). No significant associations were found between the β-diversity estimates (using the Bray–Curtis distance (Figure 2(a2), H = 1.8, *p* = 0.233) and weighted UniFrac distance (Figure 2(b2), H = 1.8, *p* = 0.233)) and the PNS cluster (low vs. high) in the longitudinal analysis.

### 2.4. Relative Abundance Analysis of the Gut Microbiome and PNS

The longitudinal relative abundance analyses showed that a higher *Bacteroidetes/Firmicutes* ratio was significantly associated with higher PNS (continuous) score over time (*p* = 0.027). At the phylum level, a higher abundance in *Bacteroidetes* was associated with higher PNS over time (*p* = 0.015) and a lower *Firmicutes* was linked to higher PNS over time (*p* = 0.037). At the genus level, *Anaerofustis*, *Tyzzerella*, *Intestinimonas*, and Family *XIII AD3011* group were positively associated with the PNS, while *Ruminiclostridium*, *Phascolarctobacterium*, and *Subdoligranulum* were inversely associated with the PNS (all *p* < 0.05).

The linear discriminant analysis (LDA) effect size (LEfSe) approach was used to compare taxa differences between low and high levels of PNS cluster (Figure 3) controlling timepoint. With an LDA score that exceeded a threshold of 2.0, we obtained relatively-consistent results as our above-described longitudinal abundance analysis. After adjusting for the time (pre-RT vs. post-RT), both the cladogram (Figure 3b) and LDA bar graph (Figure 3a,c) showed that the high PNS cluster was more abundant with one phylum (*Bacteroidetes*), one order (*Bacteroidales*), one class (*Bacteroidia*), and four genera (*Ruminiclostridium9, Tyzzerella, Eubacterium_fissicatena* group, and *DTU089*) compared to the low PNS cluster, while the low PNS cluster was more abundant with one family (*Acidaminococcaceae*) and three genera (*Lactococcus, Phascolarctobacterium*, and *Desulfovibrio*) compared to the high PNS cluster.

### 2.5. Functional Pathway Analysis of the Gut Microbiome

Using the Tax4Fun package, 280 pathways in KEGG level 3 were significantly associated with PNS and these pathways were clustered to 41 level 2 clusters of pathways according to KEGG annotations (Appendix A). After adjusting by the Benjamini–Hochberg method, Figure 4 presents the significantly-different functional pathways associated with the PNS cluster using Wilcoxon rank test, including the nervous system (*p* = 0.0719), lipid metabolism (*p* = 0.0469), glycan biosynthesis and metabolism (*p* = 0.0124), and the digestive system (*p* = 0.0216). Figure 4a presents all the 41 pathways based on KEGG annotations; Figure 4b presents 31 pathways based on KEGG annotations which only use mean abundance larger than 5% (10 levels deleted).

Based on the species level analysis with Tax4fun, using the Mann–Whitney U test, we found that glycan biosynthesis and metabolism (glycosaminoglycan degradation, *p* = 0.022; glycosphingolipid biosynthesis—globo series, *p* = 0.032; glycosaminoglycan biosynthesis-chondroitin sulfate, *p* = 0.032), vitamins (folate biosynthesis, *p* = 0.007; lipoic acid metabolism, *p* = 0.032) were significantly associated with the PNS cluster pre-RT. Additionally, endocrine system (adipocytokine signaling pathway, *p* = 0.015), lipid metabolism (fatty acid elongation, *p* = 0.046; steroid hormone biosynthesis, *p* = 0.046), immune system (RIG-I-like receptor signaling pathway, *p* = 0.046; hematopoietic cell lineage, *p* = 0.022), and nervous system (GABAergic synapse, *p* = 0.004; glutamatergic synapse, *p* = 0.010) were significantly associated with the PNS cluster pre-RT. Additionally, glycan biosynthesis and metabolism (lipopolysaccharide biosynthesis, *p* = 0.008) and vitamins (folate biosynthesis, *p* = 0.008; lipoic acid metabolism, *p* = 0.028) were significantly associated with the PNS cluster post-RT. 

## 3. Discussion

To the best of our knowledge, this is the first study to explore associations between the diversity, composition, and potential function of the gut microbiome (based on KEGG pathways) with the PNS cluster (an average score of pain, fatigue, depressive symptoms, sleep disturbance, and cognitive dysfunction) among cancer patients. This study had several noteworthy findings. The α- (e.g., evenness in particular) and β-diversity (e.g., weighted UniFrac distance) and taxa abundance of the gut microbiome were associated with the PNS clusters. Although bacterial composition of the gut microbiome has significant effects on the PNS cluster on its own, it is primarily the related microbial metabolites that cause significant biological changes. We found that the glycosaminoglycan biosynthesis, endocrine and lipid metabolism, immune system, and nervous system showed significant associations with the PNS cluster in this study. All these findings could help us understand the biological mechanisms of PNS and provide some potentially-precise targets for interventions that may relieve patients’ PNS.

The microbiome–gut–brain axis model suggested a bidirectional cross-talk between the gut and brain, linking emotional and cognitive centers of the brain with peripheral intestinal function [25]. Cancer treatments such as chemotherapy and RT could disturb the gut microbial ecosystem and the microbiome–gut–brain axis, resulting in manifestations of PNS. Until recently, a healthy gut microbiome (e.g., more microbial diversity) has emerged as a novel target to reduce the toxicity and adverse effects of cancer therapies [26,27]. Studies in animal models suggested that dysbiotic gut microbes, especially in GF mice, can heighten cancer pain [28], fatigue [29], anxiety, depression, and cognitive dysfunction. In this study, at the phylum level, we found positive trend of association between *Bacteroidetes* and the PNS cluster, and there was a negative trend or association between *Proteobacteria, Firmicutes,* and the PNS cluster prior to RT. A positive association between *Bacteroidetes/Firmicutes* ratio and PNS was also observed pre-RT. At one-month post-RT, we detected a similarly increased trend in *Bacillales*. *Bacteroidetes/Firmicutes* ratio is an indicator of inflammation [30], further increasing the PNS in cancer treatment. Interestingly, *Bifidobacteriales* showed a decrease in patients with high levels of PNS both pre- and post-RT. *Bifidobacterium* is a genus that has been used as a conventional probiotic for treating ulcerative colitis [31], indicating a less protective environment for inflammatory response in HNCs with RT. 

Similarly, the longitudinal analysis indicated a higher *Bacteroidetes/Firmicutes* ratio (linked to inflammation [30]) was significantly associated with higher PNS score over time. In addition, seven taxa (four taxa from the phylum *Firmicutes* and three taxa from the phylum *Bacteroidetes*) were associated with the high level of PNS cluster and four taxa (three taxa from the phylum *Firmicutes* and one taxa from the phylum *Proteobacteria*) were associated with the low level of PNS cluster. Among these significant taxa associated with a high PNS, *Bacteroidetes, Bacteroidales* [32], and Bacteroides [33,34] showed close associations with inflammation response. Consistent with our data, a higher abundance of *Tyzzerella* and *Ruminiclostridium* were reported in inflammation-associated cancer [35]. Therefore, the gut microbiome may shape the severity of the PNS cluster through key mechanisms associated with immunomodulation, reduction of diversity, and ecological variability [26]. Among these taxa associated with the low PNS, *Phascolarctobacterium* and *Subdoligranulum* can produce short chain fatty acids (SCFAs) that may help suppress inflammation [36,37]; therefore, the association between lower *Phascolarctobacterium* and *Subdoligranulum* with higher PNS may indicate an increased inflammatory response for PNS in our population. An increasing inflammatory response could be activated by cancer and cancer treatments such as chemotherapy and RT. Our findings support the microbiome-gut-brain axis model as a potential framework to understand the biological role of the gut microbiome in PNS.

Although it is necessary to establish the composition of the gut microbiome, it is more important to understand the functional profiles from the metagenomic 16S rRNA data. According to the microbiome–gut–brain axis, SCFAs are key microbial metabolome pathways affecting PNS [38] through histone deacetylases, hormonal, and immune pathways [39]. Microbial regulation of tryptophan is another focus of the metabolome pathway [40]. Tryptophan can be synthesized by bacterial phyla including *Proteobacteria, Actinobacteria, and Firmicutes* [41]. A dysbiotic gut microbiome (due to cancer and cancer treatments) can dysregulate the biosynthesis and metabolism of these metabolites. Consistent with previous literature, we found that the gut microbiome can impact the PNS cluster through affect-specific metabolic pathways, including glycan metabolism, the nervous system [42], and immune and endocrine pathways [43,44,45,46]. Specifically, microbial fermentation transforms a variety of indigestible glycans into SCFAs [47]. A dysbiotic gut microbiome may affect the volume of SCFAs and therefore lead to a higher PNS level. Additionally, cytokine molecules produced in the gut can travel to the brain and activate the release of cortisol through the hypothalamic–pituitary–adrenal (HPA) axis, a further potent activator of the stress system and higher PNS [25]. In this study, the microbial function analysis via KEGG does not depict an actually-determined impact of the microbiome on the signaling/functional pathways, but merely delivers a prediction of the probability of a certain function based on the known effects of the taxa that are investigated. Additionally, the diet composition has potentially direct influence on the gut microbiome and the microbiome–gut–brain axis [48,49]. Specifically, a Western diet rich in protein or fat may lead to significant reductions in *Bifidobacteria* and butyrate-producing bacteria [50,51,52]. In contrast, supplementation of high fat diet with high fibers has shown to reverse the depleted levels of *Bifidobacteria* and butyrate-producing bacteria [53]. Thus, the gut microbiome has been proposed as one important component in nutritional intervention studies to improve mental health outcomes such as anxiety and depression [54]. Patients with cancer reported malnutrition and changes in diet composition [55], which may adjust the relationship between gut microbiome and PNS. As a limitation of this study, future work should consider diet in data analysis. Nevertheless, our findings provide critically important clinical data to understand the functions of the gut microbiome in cancer. Taken together, findings of this study seem to corroborate the important role of the gut microbiome in the PNS cluster via several pathways associated with microbiome–gut–brain axis. 

This study has several limitations. We analyzed 13 patients pre- and post-RT from a single research institution in the Southeast region of the United States. The small sample size and single institution for enrolment may limit the generalizability of our findings, and interpretation of these findings should be done with caution. Current evidence is scarce in cancer, however, this study provides promising information regarding the associations between the gut microbiota and PNS. Further, to increase the generalizability of these findings, this study should be repeated in a larger, nationwide sample. Moreover, the gut microbiome data in this study were processed and sequenced using the 16S rRNA V3-V4 gene region, whose domain is restricted to bacteria and archaea but we did not study the gut microbiome in species or strain levels [56,57]. The phylum and genus levels of the gut microbiome could not provide specific resolution to understand the function and cause–effect of microbes on PNS. Although the functional pathway analysis based on KEGG predicted the potential function of the gut microbiome, shotgun metagenomics methods are strongly recommended in future work. Lastly, diet and antibiotic usage (only two samples) were not controlled in the analysis. Future work should replicate our findings controlling for diet status and excluding samples with antibiotic usage.

## 4. Materials and Methods

### 4.1. Design

We conducted a pilot study using a longitudinal, prospective study design. All data were collected pre- and one-month post-intensity-modulated radiation therapy (IMRT). Patients were enrolled at the Oncology Clinics of Emory’s Winship Cancer Institute between September 2017 and April 2018. The study was approved by Emory University’s institutional review board (IRB00070176) and all enrolled patients provided informed consent. 

### 4.2. Sample

Inclusion criteria for all participants included being ≥ 21 years old and with a histological proof of squamous cell carcinoma of the head and neck region. In addition, patients had to be diagnosed with clinical cancer stage T1-4, any N with no distant metastasis, scheduled to receive IMRT, and had adequate organ function. Patients with simultaneous primary tumors, those with previous invasive malignancies less than three years prior to enrollment, and pregnant women were excluded. Additional exclusion criteria for patients included major psychiatric disorders, chronic medical conditions involving the immune system (e.g., HIV, hepatitis B or C), regular use of immunosuppressive medications (such as glucocorticoids and methotrexate), and those who cannot speak English. For medical pilot studies, a sample size between 10 and 30 is recommended with adequate ability to test hypotheses [58]. Thirteen participants were enrolled in this study.

### 4.3. Measures

Demographic and clinical variables: Demographic variables included age, gender, race (White vs. non-White), marital status (married vs. single), history of smoking (yes vs. no), and history of alcohol use (yes vs. no). Clinical variables included body mass index (BMI), antibiotic usage (yes vs. no), comorbidities, primary cancer site (oropharynx vs. non-oropharynx), cancer stage, radiation dose, cancer treatment (radiation + surgery; radiation + chemotherapy; radiation + chemotherapy + surgery), and HPV status (positive vs. negative). The HPV status was collected via the pathology report (primarily based on p16 status using immunohistochemistry) in medical record. These variables were chosen due to their potential impacts on the PNS cluster and the gut microbiome [5,59,60,61,62].

Psychoneurological symptoms (PNS) cluster: Symptoms in the PNS cluster were measured using the Patient-Reported Outcomes version of the Common Terminology Criteria for Adverse Events (PRO-CTCAEs). The PRO-CTCAEs was developed by the National Cancer Institute to improve the accuracy and precision of clinician-reported CTCAEs [63] and has been validated for use in HNCs [64]. Symptom severity of individual PRO-CTCAE items are scored using a 5-level verbal descriptor scale and are coded from 0 to 4, with 0 indicating no symptoms and 4 suggesting severe symptoms. In this study, an average score of the five symptoms—pain, fatigue, depressive symptoms, sleep disturbances, and cognitive dysfunction were computed as the total score of the PNS cluster. As an exploratory work, the mean value of total score was used as the cutoff point for high (above mean) vs. low (below mean) PNS clusters.

Gut microbiome: Fecal specimens were collected to assess the gut microbiome. Based on the Human Microbiome Project (HMP) protocol [65], eligible patients were instructed to use the home-based stool specimen collection kit (i.e., one toilet basin (Fisher Scientific Co. L.L.C., Pittsburgh, PA, USA), one pair of gloves, and one biohazard bag with three small fecal collection tubes (Fisher Scientific Co. L.L.C., Pittsburgh, PA, USA)) to obtain stool samples. A reminder for stool sample collection was sent seven days before patients’ clinical visit. All stool samples were self-collected at home and immediately frozen in a freezer by patients before the clinical visits. On the day of the clinical visits, patients wrapped the frozen stool samples with ice packs and delivered the frozen fecal samples to research staff according to the storage protocol. The samples were immediately stored at −80 °C until DNA extraction and sequencing. 

### 4.4. DNA Extraction and Sequencing

Based on the HMP standard operating protocol, the microbial DNA was extracted from fecal samples using the PowerSoil isolation kit (MO BIO Laboratories, Carlsbad, CA, USA). In this study, 16S rRNA V3-V4 gene regions [66,67] were extracted and sequenced. The 16S libraries were prepped from 12.5 ng of DNA using the 16S metagenomic sequencing library preparation workflow. The 16S rRNA amplicons were generated using KAPA HiFi HotStart ReadyMix (KAPA Biosystems, KK2600) and primers specific to 16S V3-V4 region of Bacteria 341F (5′-CCTACGGGNGGCWGCAG-3′)-805R (5′-GACTACHVGGGTATCTAATCC-3′). PCR clean-up was performed using AMPure XP beads (Beckman, A63880) and indices were attached using the Nextera XT Index kit (Illumina, FC-131-1001). Clean-up was performed on the indexed libraries using AMPure XP beads. The 16S libraries were pooled in equal amounts based on fluorescence quantification. Each run included a control template to test for PCR accuracy and possible contamination. Final library pools were quantitated via qPCR (Kapa Biosystems, catalog KK4824). The pooled library was sequenced on an Illumina miSeq using miSeq v3 600 cycle chemistry (Illumina, catalog MS-102-3003) at a loading density of 8 pM with 20% PhiX, at PE300 reads. This process was completed at the Emory Integrated Genomics Core (https://www.cores.emory.edu/eigc/). In this study, three control samples (one non-template control [NTC] and two Zymo samples) were used. The microbial sequencing led to paired-end sequences.

### 4.5. Statistical Analysis

Descriptive statistics, including mean (standard deviation (SD)), median (range), and frequency (%) were performed for demographic and clinical variables. After sequencing the gut microbiome data, the average sequence counts per sample were 280,088 (ranging from 204,113 to 976,098). The sequence counts of all samples ranged from 204,113 to 293,050 except one sample with sequence counts of 976,098, which should be associated with bacterial load. The sequence data were trimmed at 20–275 base pairs to obtain high-quality sequences according to the Phred score of 20 (99.95% accuracy) analyzed by using DADA2. Interactive quality plots were created to visualize Phred scores for both forward and reverse sequences. Quality control of paired-end microbial sequences was implemented using DADA2 [68]. DADA2 was used to infer exact sequence variants (ASVs). An average 152,235 ± 83,343 high-quality sequences per sample were retained after quality control. Taxonomies of the gut microbiome were assigned using the pre-trained classifier on Silva 132 databases with a 99% threshold [69]. The rarefaction was conducted to normalize the data based on a sample sequencing depth in our analysis. This sequencing depth was chosen for rarefaction so that if the total count for any sample(s) were smaller than this value, those samples could be dropped from the diversity analysis. The phylogenetic tree and analysis of the gut microbiome diversity and composition were conducted using Quantitative Insights Into Microbial Ecology (QIIME) 2 [70]. The α-diversity (richness and evenness within samples) was calculated using standardized estimates—Chao1, Shannon’s index, Faith’s phylogenetic diversity (Faith_PD), and Pielou’s evenness (Pielou_e). Spearman correlational analysis was used to compute α-diversity estimates for continuous variables such as the PNS cluster; the Kruskal–Wallis test (non-parametric) was conducted to analyze α-diversity estimates in categorical variates (low vs. high PNS clusters). Bray–Curtis distance, weighted UniFrac distance metrics, and principal coordinates analysis (PCoA) were used to analyze and visualize patterns of β-diversity (dissimilarities between samples). The 2-dimensional PCoA was performed by QIIME2R using default parameters [70]. Each axis of PCoA has an eigenvalue whose magnitude indicates the amount of variation in that axis. The proportion of a given eigenvalue to the sum of all eigenvalues reveals the relative importance of each axis [71,72]. The confidence ellipse represents 95% of confidence level. Distance matrices were used to conduct permutational univariate and multivariate non-parametric analysis of dissimilarities (ADONIS) in QIIME 2. The dispersion of the variance was homogenous in both PNS groups (F = 0.343, *p* = 0.564) based on the betadisper analysis using permutation test for homogeneity of multivariate dispersions in R. The longitudinal impact of time on the gut microbiome diversity and its association with the PNS cluster was analyzed using linear mixed effect models (for the α-diversity) and pairwise distance comparisons (for the β-diversity) using QIIME q2-longitudinal plugin [73]. The mixed effect model included time, PNS, and interaction between time and PNS. 

The linear discriminant analysis (LDA) effect size (LEfSe) [74] is an algorithm for biomarker discovery and identifies genomic features (e.g., genes or taxa) characterizing the differences between multiple biological conditions. LEfSe emphasizes both statistical significance and biological relevance. In this algorithm, the Kruskal–Wallis sum–rank test is used to detect features with significant differential abundance between biological conditions; biological significance is further investigated using a set of pairwise tests among subclasses using the Wilcoxon rank-sum test. Lastly, LEfSe uses LDA to estimate the effect size of each differentially abundant feature. In our study, differences in relative abundances of bacterial taxa between high vs. low PNS clusters were identified using the LEfSe approach [74]. The Kruskal–Wallis sum–rank test detected features that had significantly-different abundances between low and high PNS clusters, while controlling time as a covariate. Then, LDA was performed to estimate the effect size of each feature (*p* < 0.05 and LDA score/effect-size threshold = 2). Time was considered as a subclass in the LEfSe analysis. 

Functional profiles of 16S rRNA gene were identified using R package Tax4Fun [75]. Tax4Fun can analyze the functional genes of bacterial communities based on the 16S rRNA sequencing data and provide a good approximation to the gene profiles obtained from metagenomic shotgun sequencing methods. This analysis was conducted in three steps: 1) The SILVA-based 16S rRNA OTU table was transformed to a taxonomic profile of the prokaryotic Kyoto Encyclopedia of Genes and Genomes (KEGG) organisms using a precomputed association matrix, 2) the estimated abundances of KEGG organisms were normalized by the 16S rRNA copy number obtained from the NCBI genome annotations, and 3) the normalized taxonomic abundances were used to combine the precomputed functional profiles of the KEGG organisms for the prediction of the functional profile of the gut microbial community [75]. Organism-specific functional profiles were computed for all bacterial genomes in KEGG. The Benjamini–Hochberg method [76] was used for multiple testing corrections. The KEGG pathways from Tax4Fun using the species levels were further analyzed using the Mann–Whitney U test to assess the differences between high vs. low PNS clusters over time. All the analyses were conducted using QIIME 2 [70,77,78] and R 3.3.3 (https://cran.r-project.org/bin/macosx/). The statistical significance level was set at *p* < 0.05.

## 5. Conclusions

In summary, the α- and β-diversity and taxa abundance showed potential associations with the PNS cluster in patients with HNCs. The high level of PNS cluster was more abundant with phylum *Bacteroidetes*, order *Bacteroidales*, class *Bacteroidia*, and four genera (*Ruminiclostridium9, Tyzzerella, Eubacterium_fissicatena* group, and *DTU089*), while the low level PNS group was more abundant with family *Acidaminococcaceae* and three genera (*Lactococcus, Phascolarctobacterium*, and *Desulfovibrio*). Glycan biosynthesis and metabolism (lipopolysaccharide biosynthesis) and vitamins (folate biosynthesis and lipoic acid metabolism) seemed to predict the levels of the PNS cluster across time. In brief, diversity and taxa abundance of the gut microbiome play a promising role in understanding the biological mechanisms of PNS among HNCs and our findings should be tested in a large and multi-site sample.

## Figures and Tables

**Figure 1 cancers-12-02531-f001:**
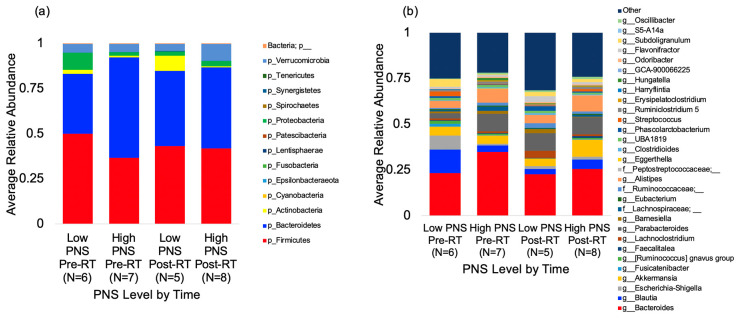
Gut microbial taxa at the phylum (**a**) and genus (**b**) level by time (pre-RT vs. post-RT) and PNS levels (low vs. high). All the microbial phyla and top 20 genera are displayed. PNS, psychoneurological symptoms; RT, radiation therapy.

**Figure 2 cancers-12-02531-f002:**
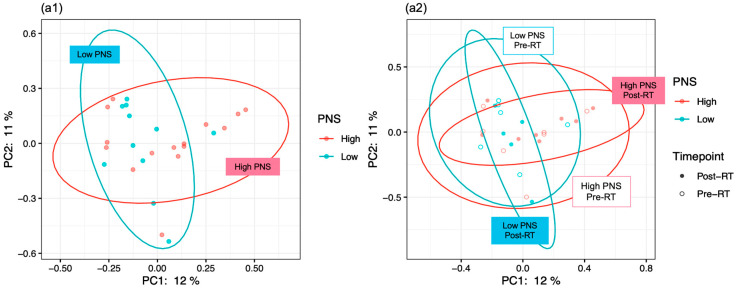
The β-diversity of gut microbiome by PNS level (low vs. high) pre-RT, post-RT, or the combination of both. (**a1**) and (**b1**) display the dissimilarities of gut microbiome via PNS levels (low vs. high) using the Bray-Curtis and weighted UniFrac distance. (**a2**) and (**b2**) display the dissimilarities of gut microbiome via PNS levels (low and high) vs. timepoint (pre-RT and post-RT) using Bray-Curtis and weighted UniFrac distance. PNS, psychoneurological symptoms; RT, radiation therapy.

**Figure 3 cancers-12-02531-f003:**
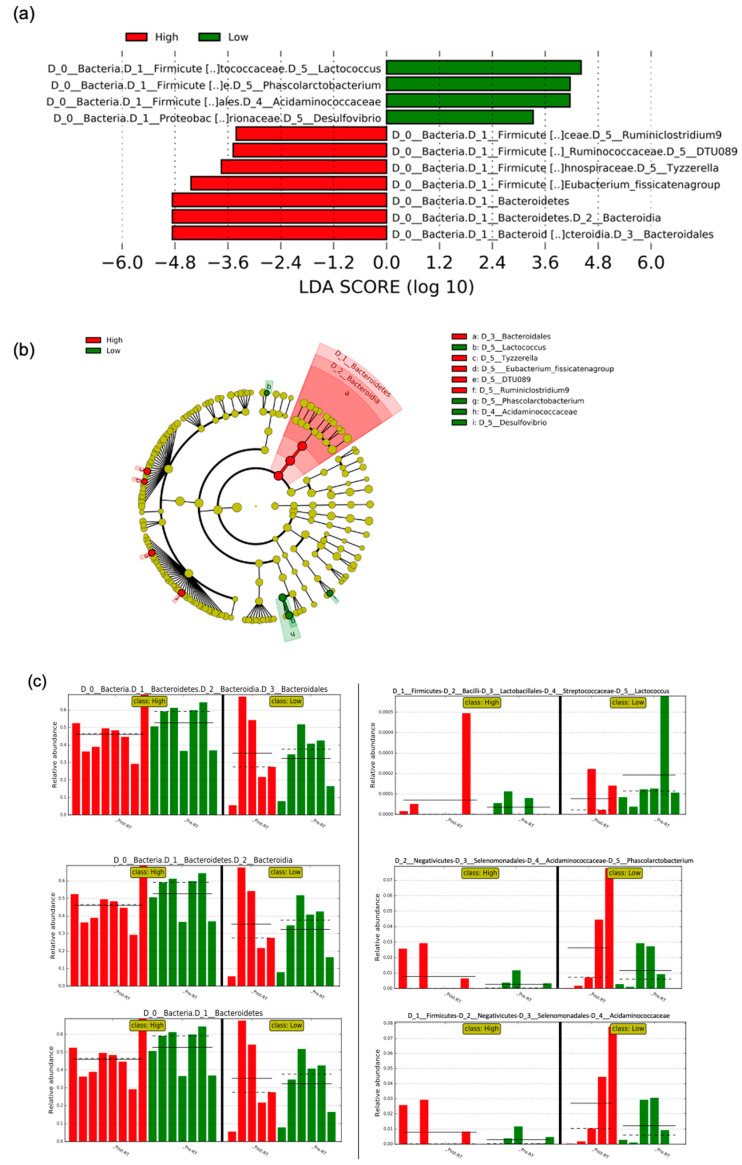
Linear discriminant analysis (LDA) effect size (LEfSe) for the gut microbial abundance based on PNS levels (low vs. high). (**a**) shows the overabundant taxa for low vs. high PNS, (**b**) shows the cladogram of taxa associated with PNS, and (**c**) shows the three most significant biomarkers of high (left panel) vs. low (right panel) PNS by time (pre-RT vs. post-RT); the solid straight lines mean plot subclass means; the dotted straight lines mean plot subclass medians. Low indicates low PNS level; High indicates high PNS level. PNS, psychoneurological symptoms; RT, radiation therapy.

**Figure 4 cancers-12-02531-f004:**
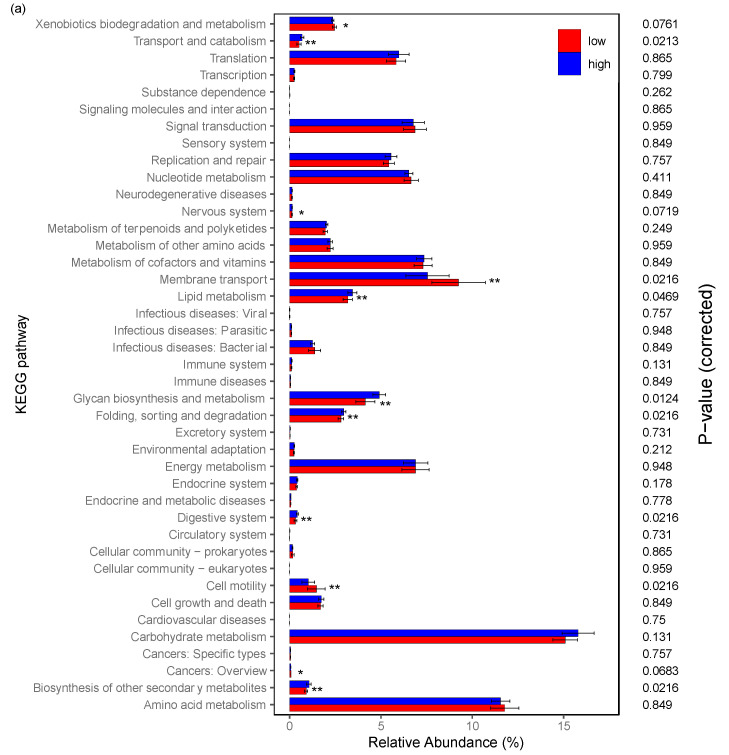
Functional pathway analysis of the gut microbiome based on PNS level (low vs. high). Based on the KEGG pathway, significant metabolism pathways are presented adjusting by the Benjamini–Hochberg method. (**a**) indicates all the 41 pathways based on KEGG annotations and (**b**) presents 31 pathways based on KEGG annotations using mean abundance larger than 5%. PNS, psychoneurological symptoms; KEGG, Kyoto Encyclopedia of Genes and Genomes. * *p* < 0.10; ** *p* < 0.05.

**Table 1 cancers-12-02531-t001:** Demographic and clinical characteristics (*n* = 13).

Variables	N (%)
**Age, years**	
Mean (SD); median (range)	60.0 (9.4); 57.0 (47.0–76.0)
**Men**	11 (84.6)
**White**	11 (84.6)
**Alcohol status, drinks per week**	
<1	4 (30.8)
≥1	9 (69.2)
**Smoking status**	
Ever	10 (76.9)
Never	3 (23.1)
**Married**	8 (61.5)
**BMI**	
Mean (SD); median (range)	26.0 (4.7); 26.0 (18.3–35.2)
**Cancer site**	
Oropharynx	6 (46.2)
Non-oropharynx	7 (53.8)
**Positive HPV status**	7 (63.6)
**Cancer stage**	
II	2 (15.4)
III	3 (23.1)
IV	8 (61.5)
**Treatment modality**	
Radiation + surgery	6 (46.2)
Radiation + chemo	5 (38.5)
Radiation + chemo + surgery	2 (15.4)
**Received surgery**	8 (61.5)
**RT dose**	
Mean (SD); median (range)	64.9 (5.3); 66.0 (54.0–70.0)
**Antibiotics use ^a^**	
Yes	1 (7.7)
No	11 (84.6)
**Anti-inflammatory drugs use ^a^**	
Yes	5 (38.5)
No	7 (53.0)
**Continuous PNS score pre-RT**	
Mean (SD); median (range)	6.8 (3.4); 6.5 (1.5–13.0)
**Continuous PNS score post-RT**	
Mean (SD); median (range)	6.7 (4.4); 6.5 (1.0–19.0)
**Categorical PNS cluster pre-RT**	
High vs. low	8 (61.5%) vs. 5 (38.5)
**Categorical PNS cluster post-RT**	
High vs. low	7 (53.8) vs. 6 (46.2)

^a^ Missing value: antibiotics usage with one missing value; anti-inflammatory with one missing value. BMI, body mass index; chemo, chemotherapy; HPV, human papillomavirus; PNS, psychoneurological symptoms; RT, radiation therapy; SD, standard deviation.

**Table 2 cancers-12-02531-t002:** Longitudinal association between the gut microbial α-diversity and PNS by time (*n* = 13).

Variable	Faith_PD			Evenness		
	Coefficient	z	*p*	Coefficient	z	*p*
Time	1.601	3.603	<0.0001	−0.033	−1.495	0.135
PNS cluster	0.365	0.409	0.683	−0.039	−2.233	0.026
Time*PNS cluster	−1.101	−4.413	<0.0001	0.009	0.765	0.445

Time*PNS cluster means interactions between Time and PNS cluster; Faith_PD, Faith’s phylogenetic diversity; PNS, psychoneurological symptoms.

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
