# Peer review of "Gut Microbiome Associated with the Psychoneurological Symptom Cluster in Patients with Head and Neck Cancers"

_cancers, 2020, doi:10.3390/cancers12092531_

Round 1

Reviewer 1 Report

The authors have investigated the linkage between diversity of the gut microbiome and the occurrence/severity of psychoneurological symptoms (PNS) in the post-radiation therapy phase. To this purpose, pre- and post-radiation therapy specimens of stool were analyzed by 16S DNA sequencing to determine the diversity of the gut microbiome in different stages of therapy.

The longitudinal relative abundance analyses showed that the representation level of the most phila was either directly or inversely associated with either of the PNS (continuous) scores over time. To pinpoint the precise linkages, LEfSe approach was used to compare taxa differences between low and high levels of PNS.

Linkage between bacterial abundancies and the PNS clusters , subsequent functional pathway analyses of 16s RNA pre- and post-therapy were performed. Microbiomes of the high- and low PNS cluster patients exhibited differential abundance at a taxonomic level  of phila and families, ending in process divergence of glycan and vitamin metabolism (Lipopolysaccharide biosynthesis, folate biosynthesis and lipoic acid metabolism).

The authors interpret this linkage between diversity, composition, and the predicted function of gut microbiome with the PNS cluster as a probable impact of the gut microbiome diversity and abundance on the development of PNS in cancer patients, post-therapeutically.

The causality between the microbiome diversity/abundance and the PNS has been introduced in the Discussion section based on the existing evidence, connected with the data generated in this study.

Comments

The manuscript has been written clearly and in a proper scientific manner, few remarks on definitions and sentence structure aside.

The authors are addressing the main study limitations. N of 13 patients, even reduced to a N=2 for Group Radiation + chemo + surgery is not sufficient for the  generalizability of the findings, therefore the data produced exert a limited impact . This low N is the main concern when it comes to the reliability of the data in this study.

Methods

The methodology for obtaining the experimental data and for their analysis described herein has been correctly implemented.

The major limitation is presented by a low sample (N=13).

The sample contains a bias in terms of gender, ethnicity and lifestyle. Nevertheless, since no major differences have been determined between these subgroups in the patient sample, as stated by the authors, we can observe it as non-biased. It would be good if the authors corroborated this statement some more.

General remark on Methods: it would be useful if the more complex statistical analysis methods were very shortly described (purpose, type of test used, parametric/non parametric, distribution, output, confidence interval). In complex software packages, the principle and the tests that the package employs.

Remark: Using the Tax4Fun package, Figure 4 presents the significantly different functional pathways associated with the PNS cluster in the second level of KEGG,…

Please define the basics of the significant difference of the fuctional pathways, which categories, which parameter is different; generally, it is about the KEGG pathways that were predicted as significantly more active based on the impact of the determined microbiome taxa?

Please explain in short the principle of the Tax4Fun package. How does work and how does it align with the KEGG data?

Results: Second sentence in the paragraph 2.5 Functional Pathways Analysis of the Gut Microbiome is extremely long and difficult to follow, please rephrase or set into separate sentences.

The following sentence should be precisely re-formulated: 2.3 The PNS cluster (low vs. high) had no significant difference in Bray-Curtis distance (Figure 2a,

Please complete the depictions ‘high’ and ‘low’ in the figure 3

Discussion

Although the limitation of the KEGG Analysis has been mentioned, it would be necessary to intensify this statement. The analysis by Kyoto Encyclopedia of Genes and Genomes (KEGG) is an online tool that allows an in silico generation of prediction data based on the existing, previously published data and the data from the study. It does not depict an actually determined impact of the microbiome on the signaling/functional pathways, but merely delivers a prediction of the probability of a certain function based on the known effects of the taxa that are investigated.

Also, study restriction of the gut microbiome in species or strain levels should be explained in more detail (what are the consequences of this particular restriction?). Since the main differences found between the outcome groups +PNS and –PNS were determined on the level of phylum and family, restrictions at a taxonomic  level lower than species appear to be acceptable. Please introduce a comment on this in the discussion.

Author Response

Reviewer 1

The authors are addressing the main study limitations. N of 13 patients, even reduced to a N=2 for Group Radiation + chemo + surgery is not sufficient for the generalizability of the findings, therefore the data produced exert a limited impact. This low N is the main concern when it comes to the reliability of the data in this study.

Response: We admitted the small sample size as major limitation in this study. As a pilot study, this small sample size provides important information and we have addressed the generalization of our findings should be cautious, see the limitation section.

Methods

The major limitation is presented by a low sample (N=13).

Response: Agreed and we have addressed this in our limitation section. Given current evidence is scarce in cancer, this study provides promising information regarding associations between the gut microbiota and PNS.

The sample contains a bias in terms of gender, ethnicity and lifestyle. Nevertheless, since no major differences have been determined between these subgroups in the patient sample, as stated by the authors, we can observe it as non-biased. It would be good if the authors corroborated this statement some more.

Response: The study population represented the current population in HNCs as cited in other publications.

General remark on Methods: it would be useful if the more complex statistical analysis methods were very shortly described (purpose, type of test used, parametric/non parametric, distribution, output, confidence interval). In complex software packages, the principle and the tests that the package employs.

Response: We have added detailed information for all the complex statistical analysis methods such as LEfSe and Tax4Fun.

Remark: Using the Tax4Fun package, Figure 4 presents the significantly different functional pathways associated with the PNS cluster in the second level of KEGG,… Please define the basics of the significant difference of the functional pathways, which categories, which parameter is different; generally, it is about the KEGG pathways that were predicted as significantly more active based on the impact of the determined microbiome taxa?

Response: The functional pathways in KEGG were determined based on the Silva-based 16S rRNA OTU table as described in the statistical analysis. Then we analyzed how the PNS level associated with these KEGG pathways at levels 3 and 2. We reported level 2 in Figure 4. We revised the statistical analysis and results sections.

Please explain in short the principle of the Tax4Fun package. How does work and how does it align with the KEGG data?

Response: We added more information about this in the statistical analysis.

Results: Second sentence in the paragraph 2.5 Functional Pathways Analysis of the Gut Microbiome is extremely long and difficult to follow, please rephrase or set into separate sentences.

Response: Revised

The following sentence should be precisely re-formulated: 2.3 The PNS cluster (low vs. high) had no significant difference in Bray-Curtis distance (Figure 2a,….

Response: Revised

Please complete the depictions ‘high’ and ‘low’ in the figure 3

Response: We noted the meaning of low and high in the Figure legend.

Discussion

Although the limitation of the KEGG Analysis has been mentioned, it would be necessary to intensify this statement. The analysis by Kyoto Encyclopedia of Genes and Genomes (KEGG) is an online tool that allows an in silico generation of prediction data based on the existing, previously published data and the data from the study. It does not depict an actually determined impact of the microbiome on the signaling/functional pathways, but merely delivers a prediction of the probability of a certain function based on the known effects of the taxa that are investigated.

Response: Thanks for this comment. We have added more information regarding this in the limitation section.

Also, study restriction of the gut microbiome in species or strain levels should be explained in more detail (what are the consequences of this particular restriction?). Since the main differences found between the outcome groups +PNS and –PNS were determined on the level of phylum and family, restrictions at a taxonomic level lower than species appear to be acceptable. Please introduce a comment on this in the discussion.

Response: We explained more of this in the limitation section and thanks.

Reviewer 2 Report

This article by Jinbing Bai et al. Describe very interesting results about the modulation of the gut microbiome regarding the PNS in context of head and neck cancers. Some modification and reviewing of the manuscript have to be applied before possible publication.

Major :

The library sequencing method (V3-V4 region at least) has to be detailed in the abstract.

As discussed by the authors, the study is monocentric and limited in size. Have they determined the needed number of patient to include? If yes, describe it. If not, why ? Moreover, the period of inclusion has to be detailed, as a seasonal modification of the microbiome could bias results.

Has the investigators collect details on the cancer stage, why haven't they stratified their results accordingly ?

It is pretty surprising that the antibiotic usage was not considered as an exclusion factor. Could the authors justify their choice and give detail about the “antibiotic usage (yes vs. no)” criteria?

How was determined the HPV status ? Give details.

More precision have to be given about the collection of the sample (nature of the sampling tube, delay between sampling and collection, method to freeze the sample, ...).

More precision have to be given to justify the choice of the V3-V4 region amplified for the gut microbiome analysis, as the authors stated on the possible limit induced by this choice. Furthermore, details have to be given (or reference) about the 16S PCR (in house PCR? Commercial PCR? Number of cycle?...), including the nature of the control template (collection sample ? PCR-free water? Other stool sample?)

Minor:

Beware of the inappropriate use of the bold and italic police. Moreover, values below eleven have to be written in full letter.

Suppl. Figure could be fused with Figure 1 to present all their results at once.

Author Response

Reviewer 2
Major:
The library sequencing method (V3-V4 region at least) has to be detailed in the abstract.

Response: We added this in the abstract.
As discussed by the authors, the study is monocentric and limited in size. Have they determined the needed number of patient to include? If yes, describe it. If not, why? Moreover, the period of inclusion has to be detailed, as a seasonal modification of the microbiome could bias results.

Response: We added how the sample size was estimated for a pilot study and also the data collection period.
Has the investigators collect details on the cancer stage, why haven't they stratified their results accordingly ?
Response: This information is in Table 1. We did not stratify the results because of no difference in alpha-diversity.
It is pretty surprising that the antibiotic usage was not considered as an exclusion factor. Could the authors justify their choice and give detail about the “antibiotic usage (yes vs. no)” criteria?

Response: In this study, only one patient received antibiotics pre-RT and one post-RT during the data collection period. We added the information in the demographic data.
How was determined the HPV status? Give details.
Response: The HPV was determined by the pathology report in medical record.
More precision have to be given about the collection of the sample (nature of the sampling tube, delay between sampling and collection, method to freeze the sample, ...).
Response: More information was added in the methods for stool sample collection.
More precision have to be given to justify the choice of the V3-V4 region amplified for the gut microbiome analysis, as the authors stated on the possible limit induced by this choice. Furthermore, details have to be given (or reference) about the 16S PCR (in house PCR? Commercial PCR? Number of cycle?...), including the nature of the control template (collection sample ? PCR-free water? Other stool sample?)
Response: References were cited for V3-V4 choice for gut microbiome. The PCR and sequencing were conducted in Emory Integrated Genomics Core as added. We added related information for 3 controls (1 NTC and 2 Zymo).
Minor:
Beware of the inappropriate use of the bold and italic police. Moreover, values below eleven have to be written in full letter.

Response: Revised
Suppl. Figure could be fused with Figure 1 to present all their results at once.

Response: As Figure 1 and Supplement Figure displayed the similar information, we removed Supplement Figure after discussion with co-authors.

Reviewer 3 Report

Bai and colleagues provide a proof-of-concept study to evaluate whether the side psychoneurological effects associated with the anti-tumoral therapies in patients with head and neck cancers, are consistent with changes in composition and function of the gut microbiota. Overall, the paper is well written, and the reported associations could support future interventions targeting the gut microbiome to mitigate the side effects of the anti tumoral treatments. However, there is number of points that should be clarified before considering this manuscript for publication.

1-The number of obtained reads is highly variable (95,849-548,061). How can the authors explain this huge variability in the sequencing outcome? Authors did use high bacterial density samples (feces), which makes me think that either the DNA extraction or the library prep protocols were not optimized. Authors should provide a quantification of the bacterial load through qPCR. This would also help to support the claims make by the authors, and would provide a more realistic picture through the normalization of the semi-quantitative sequencing data using the qPCR values.

Additional questions in regards to the 16S profiling: Which primers did the authors used for targeting the V3-V4 region? If the range in the number of reads was from 85,849-548,061, what was the rationale to rarefy the community to 7,500 reads? Please also provide the MultiQC (html) report for all of the samples before being analysed with DADA2 to have an idea about the quality of the sequencing data. Please provide details onto how adapters and barcodes were removed.

2- Although it looks like that the authors did control for many known confounders of microbiome composition (antibiotics, BMI, smoking or age), they missed a key parameter that is a main driver of gut microbial diversity: diet. How can the authors assure that the reported differences in microbial composition are not confounded by this parameter?

3- Figure 2B, seems to be a fishing approach to see which beta diversity estimate provides a statistical significant PERMANOVA result. I do not think the authors have enough evidence to justify that the beta diversity is different between high and low PNS. Bray distances do not show differences, but phylogenetic distances do. However, the authors missed one biologically relevant output of the PERMANOVA results, which are the R2 values. In both analyses, the variance explained by PNS group is lower than 10%. Also, one of the PERMANOVA assumptions is that the dispersion of the variance is homogenous in both groups. This has to be addressed and stated in the manuscript.Also, it is not clear what is representing the percentage for each component of the PCoA model. I would say that is percentage of variance explained. However, since the authors are using a matrix of metric/semimetric beta diversity distances (not Euclidean and a covariance matrix) for dimension reduction, it is unclear how the magnitude of eigenvalues is related to the variance explained in each PCoA axis.

The authors performed a longitudinal evaluation (baseline-post treatment), but in my eyes they did not take advantage of this approach. Authors can calculate Procrustes distances between paired samples of the same patient (pre- post- treatment) and evaluate if those distances are different between high and low PNS groups. This would provide more clear evidence onto whether there are structural changes in the microbiota robustly associated with the neurological phenotype.

4- Figure 3. What is the contribution of each of the differential features to the overall community in both groups?

5- Figure 4. Were those p-values corrected for multiple testing? I can see that the effect size is really small. Do the authors truly believe that these small differences in functional potential of the gut microbiota could drive/be involved in the neurological side effects associated with the anti tumoral therapies?

6- Since the outcome of the analysis should change based on the criteria to include the samples in the high PNS and low PNS group, authors should provide a clear justification for including each sample in each group. They have also to provide a rationale for having two PNS group and not more than two. What was the rationale and which were the cut-offs for each category?

Author Response

Reviewer 3

1-The number of obtained reads is highly variable (95,849-548,061). How can the authors explain this huge variability in the sequencing outcome? Authors did use high bacterial density samples (feces), which makes me think that either the DNA extraction or the library prep protocols were not optimized. Authors should provide a quantification of the bacterial load through qPCR. This would also help to support the claims make by the authors, and would provide a more realistic picture through the normalization of the semi-quantitative sequencing data using the qPCR values.

Response: This number is feature frequencies after quality control. The average sequence counts per sample were 280,088 (ranging from 204,113 to 976,098) before quality control, as added in 4.5 Statistical analysis.

Additional questions in regards to the 16S profiling: Which primers did the authors used for targeting the V3-V4 region? If the range in the number of reads was from 85,849-548,061, what was the rationale to rarefy the community to 7,500 reads? Please also provide the MultiQC (html) report for all of the samples before being analysed with DADA2 to have an idea about the quality of the sequencing data. Please provide details onto how adapters and barcodes were removed.

Response: We added the primers we used in 4.4 DNA extraction. We added the sequencing counts per sample in 4.5 Statistical Analysis. The trim/truncation were at 20-275 in QIIME 2 based on Phred score of 20 (99.95% accuracy) and the trim was set to remove the primers. Adapters and barcodes have been removed in the lab. The rarefaction reads were determined based on the frequency reads of samples.

2- Although it looks like that the authors did control for many known confounders of microbiome composition (antibiotics, BMI, smoking or age), they missed a key parameter that is a main driver of gut microbial diversity: diet. How can the authors assure that the reported differences in microbial composition are not confounded by this parameter?

Response: Thanks for the great comment. Diet plays a role in the gut microbiome and we have added this as a limitation in this study.

3- Figure 2B, seems to be a fishing approach to see which beta diversity estimate provides a statistical significant PERMANOVA result. I do not think the authors have enough evidence to justify that the beta diversity is different between high and low PNS. Bray distances do not show differences, but phylogenetic distances do. However, the authors missed one biologically relevant output of the PERMANOVA results, which are the R2 values. In both analyses, the variance explained by PNS group is lower than 10%. Also, one of the PERMANOVA assumptions is that the dispersion of the variance is homogenous in both groups. This has to be addressed and stated in the manuscript. Also, it is not clear what is representing the percentage for each component of the PCoA model. I would say that is percentage of variance explained. However, since the authors are using a matrix of metric/semimetric beta diversity distances (not Euclidean and a covariance matrix) for dimension reduction, it is unclear how the magnitude of eigenvalues is related to the variance explained in each PCoA axis.

Response: We agreed with this comment regarding the power for beta-diversity difference. We have revised the conclusion based on this comment. We added that the dispersion of the variance is homogenous. We add statistical results for Figure 2 a2 and b2. The PCoA was created based on default QIIME2R software and we have cited this. PCoA represents the percentage of variance explained.

The authors performed a longitudinal evaluation (baseline-post treatment), but in my eyes they did not take advantage of this approach. Authors can calculate Procrustes distances between paired samples of the same patient (pre- post- treatment) and evaluate if those distances are different between high and low PNS groups. This would provide more clear evidence onto whether there are structural changes in the microbiota robustly associated with the neurological phenotype.

Response: The q2-longituginal pairwise-distances operates on a distance matrix to assess the distance between “pre” and “post” sample pairs, and tests whether these paired differences are significantly different between different groups. We have cited the reference from Bokulich NA, Dillon MR, Zhang Y, et al. q2longitudinal: Longitudinal and pairedsample analyses of microbiome data. mSystems. 2018b;3:e00219e318.

4- Figure 3. What is the contribution of each of the differential features to the overall community in both groups?

Response: The LEfSe concludes that specific taxa could be enriched in one group than others based on the LDA score. We may see the rank of this differential features by mean difference or significant level. This method is not to evaluate the contribution of each taxa to the overall community. We clarified the LEfSe approach in the statistical analysis.

5- Figure 4. Were those p-values corrected for multiple testing? I can see that the effect size is really small. Do the authors truly believe that these small differences in functional potential of the gut microbiota could drive/be involved in the neurological side effects associated with the anti-tumoral therapies?

Response: We did not correct the multiple testing. Even with limitations, our findings are promising as we concluded and future work should be conducted in a large sample. We revised our conclusion as well.

6- Since the outcome of the analysis should change based on the criteria to include the samples in the high PNS and low PNS group, authors should provide a clear justification for including each sample in each group. They have also to provide a rationale for having two PNS group and not more than two. What was the rationale and which were the cut-offs for each category?

Response: Mean value of PNS was used as cutoff point for low (below mean) vs. high (above mean) levels of PNS, see 4.3 Psychoneurological symptoms (PNS) cluster. As a small sample and exploratory work, we only focused on two groups. Our findings will provide promising data to design large studies.

Round 2

Reviewer 2 Report

Dear authors,

Most  of our comments were correctly addressed by your revised version. Nevertheless, I noted comments that need to be followed more extensively:

(From my initial recommandations)

Has the investigators collect details on the cancer stage, why haven't they stratified their results accordingly ?
Response: This information is in Table 1. We did not stratify the results because of no difference in alpha-diversity.

  • The absence of difference in alpha diversity could not allow conclusion for betadiversity for example. Please complete.

It is pretty surprising that the antibiotic usage was not considered as an exclusion factor. Could the Authors justify their choice and give detail about the “antibiotic usage (yes vs. no)” criteria?
Response: In this study, only one patient received antibiotics pre-RT and one post-RT during the data collection period. We added the information in the demographic data.
à As this data is crucial and could deeply impact the diversity and the specific bacterial representation in the microbiome, these patients have to be considered by doing the analyses after their exclusion.

How was determined the HPV status? Give details.
Response: The HPV was determined by the pathology report in medical record.

  • Even based on medical record, a crucial data as HPV could be impacted by the method of determination. Have the pathologists used cellular identification? Immunofluorescence? PCR? These data have to be considered.

From Other reviewers'recommandations :

5- Figure 4. Were those p-values corrected for multiple testing? I can see that the effect size is really small. Do the authors truly believe that these small differences in functional potential of the gut microbiota could drive/be involved in the neurological side effects associated with the anti-tumoral therapies?

Response: We did not correct the multiple testing. Even with limitations, our findings are promising as we concluded and future work should be conducted in a large sample. We revised our conclusion as well.

  • This limitation is crucial as, without multiple testing, results could be over-interpretated (difference could be due only to multiplication of tests). Please consider multiple test correction (qvalue could be produced using QIIME2 pipelines)

Author Response

Reviewer 2:
Has the investigators collect details on the cancer stage, why haven't they stratified their results accordingly?
Response: This information is in Table 1. We did not stratify the results because of no difference in alpha-diversity.

  • The absence of difference in alpha diversity could not allow conclusion for beta-diversity for example. Please complete.

Response: We completed this for beta-diversity and thanks for this comment.

It is pretty surprising that the antibiotic usage was not considered as an exclusion factor. Could the Authors justify their choice and give detail about the “antibiotic usage (yes vs. no)” criteria?
Response: In this study, only one patient received antibiotics pre-RT and one post-RT during the data collection period. We added the information in the demographic data.

  • As this data is crucial and could deeply impact the diversity and the specific bacterial representation in the microbiome, these patients have to be considered by doing the analyses after their exclusion.

Response: We tested all the analysis between antibiotic use vs no antibiotic use, no significant difference for alpha and beta-diversity. We also addressed this in the limitation as well.

How was determined the HPV status? Give details.
Response: The HPV was determined by the pathology report in medical record.

  • Even based on medical record, a crucial data as HPV could be impacted by the method of determination. Have the pathologists used cellular identification? Immunofluorescence? PCR? These data have to be considered.

Response: The HPV status was collected via the pathology report (primarily based on p16 status using immunohistochemistry) in medical record. We added this in 4.3 demographic and clinical variables.

From Other reviewers' recommendations: 

5- Figure 4. Were those p-values corrected for multiple testing? I can see that the effect size is really small. Do the authors truly believe that these small differences in functional potential of the gut microbiota could drive/be involved in the neurological side effects associated with the anti-tumoral therapies?

Response: We did not correct the multiple testing. Even with limitations, our findings are promising as we concluded and future work should be conducted in a large sample. We revised our conclusion as well.

  • This limitation is crucial as, without multiple testing, results could be over-interpretated (difference could be due only to multiplication of tests). Please consider multiple test correction (q-value could be produced using QIIME2 pipelines)

Response: We corrected the multiple testing in Figure 4 by the Benjamini-Hochberg method, see the update in 2.5 Functional Pathways Analysis of the Gut Microbiome.

Reviewer 3 Report

Authors have revised their manuscript and they have reasonably responded to most of my comments. Nevertheless, a number of major issues remain unaddressed.

1- the range of the reads is highly variable. The authors have to provide an explanation for this. These differences could arise from differential bacterial biomass, which have to be addressed through quantitative PCR.

Marked differences in sequencing depth could result in artificial differences between microbial profiles as a consequence of identifying “rare biosphere” in the samples in which the sequencing effort was higher (e.g. authors report that Lactococcus is a differential feature as per LefSe analysis, and this OTU represents a maximum of 0.0005 % of the overall community in one of the patients). This is of outmost importance given that the authors are not providing details on how the dataset was normalised / how they did account for differences in sequencing depth. This has to be detailed as well.

2- Based on one of my comments, authors stated in the revised version that (lines 354-355) “The dispersion of the variance is homogenous in both groups.” But no details are given in relation to which statistical test was used to reach this conclusion. Figure 2b2 suggests that the dispersion of the variance between groups is not the same (on the basis of the area of the ellipses). Homogeneity condition is a strong assumption in PERMANOVA (it is analogous to variance homogeneity in univariate ANOVA), as different spreads in the variance between groups may potentially explain the significant value obtained through the PERMANOVA test in this figure. Details on the test used have to be provided.

Following my comment on figure 2. PCoA operates using a distance-based matrix (semimetric in the case of Bray-Curtis), not a covariance matrix (as PCA) so the percentage indicated in the axes cannot represent proportion of variance explained unless some kind of transformation is done. Authors have to clearly explain this.

It looks like the authors have pasted the output of the analyses from QIIME without explaining what is being represented in the figure captions: Authors have to explain what are representing the ellipses in figure 2. They also has to describe what is being represented by the solid and dotted straight lines in Figure 3C.

3- Correction for multiple testing in Figure 4 is a must and has to be performed by the authors, as this is a way of controlling for false positives in multiple testing.

4- Not controlling for diet is a serious limitation in this study, and it deserves to be elaborated by the authors in the discussion. So far is limited to a single sentence in line 268.

Author Response

Reviewer 3

1- the range of the reads is highly variable. The authors have to provide an explanation for this. These differences could arise from differential bacterial biomass, which have to be addressed through quantitative PCR. Marked differences in sequencing depth could result in artificial differences between microbial profiles as a consequence of identifying “rare biosphere” in the samples in which the sequencing effort was higher (e.g. authors report that Lactococcus is a differential feature as per LefSe analysis, and this OTU represents a maximum of 0.0005 % of the overall community in one of the patients). This is of outmost importance given that the authors are not providing details on how the dataset was normalised / how they did account for differences in sequencing depth. This has to be detailed as well.

Response: The data processing followed standard 16S protocol based on the Human Microbiome Project in Emory Integrated Genomics Core, as reported in 4.4. Data extraction and sequencing. A rarefied table was created in the QIIME2 pipeline which was used in following analysis. This sequencing depth was chosen for rarefaction so that If the total count for any sample(s) are smaller than this value, those samples were dropped from the diversity analysis.

2- Based on one of my comments, authors stated in the revised version that (lines 354-355) “The dispersion of the variance is homogenous in both groups.” But no details are given in relation to which statistical test was used to reach this conclusion. Figure 2b2 suggests that the dispersion of the variance between groups is not the same (on the basis of the area of the ellipses). Homogeneity condition is a strong assumption in PERMANOVA (it is analogous to variance homogeneity in univariate ANOVA), as different spreads in the variance between groups may potentially explain the significant value obtained through the PERMANOVA test in this figure. Details on the test used have to be provided.

Following my comment on figure 2. PCoA operates using a distance-based matrix (semimetric in the case of Bray-Curtis), not a covariance matrix (as PCA) so the percentage indicated in the axes cannot represent proportion of variance explained unless some kind of transformation is done. Authors have to clearly explain this.

It looks like the authors have pasted the output of the analyses from QIIME without explaining what is being represented in the figure captions: Authors have to explain what are representing the ellipses in figure 2. They also has to describe what is being represented by the solid and dotted straight lines in Figure 3C. 

Response: We rechecked the dispersion of the variance between PNS clusters using scatter plot and independent sample t-test. We revised the explanation of the axis. We added info for the ellipses in Figure 2 and solid and dotted straight lines in Figure 3C.

3- Correction for multiple testing in Figure 4 is a must and has to be performed by the authors, as this is a way of controlling for false positives in multiple testing.

Response: We corrected the multiple testing in Figure 4 by the Benjamini-Hochberg method, see the update in 2.5 Functional Pathways Analysis of the Gut Microbiome.

4- Not controlling for diet is a serious limitation in this study, and it deserves to be elaborated by the authors in the discussion. So far is limited to a single sentence in line 268.

Response: We added the impact of the diet in gut microbiome in the discussion.

Round 3

Reviewer 2 Report

Dear Authors,

Thanks for your work regarding to my previous comments.

I think that the manuscript is now suitable for publication.

Author Response

Thanks so much.

Reviewer 3 Report

The authors have revised the manuscript but they have not properly addressed two critical questions I asked in the previous rounds of review:

1- Authors have to provide a better explanation to the huge variability shown in sequencing depth, rather than it was performed following standard protocols. If this is a consequence of differences in bacterial load, this has to be confirmed by the authors. I strongly recommend the quantification of the DNA extracts using qPCR.

2- Figure 2. Homogeneity in the dispersion of the variance assumption in PERMANOVA results. Since part of the claims made by the authors are based on the results of this figure, it is critical to assess that the spread of the variance assumption is met using adequate methods. Authors stated in their response and the manuscript (lines 385-386)
"The dispersion of the variance is homogenous in both PNS groups based on scatter plot and  independent sample t-test." T-test is not designed to check if two multivariate datasets are homoscedastic, but to check if there are differences between means coming from two normally distributed populations. Scatter plots are not statistical test either. If using R, authors can do this using the function betadisper provided in the package vegan. Authors have to provide the relevant values for the statistics as well as the p-values.

Also, in regards to one of my questions, authors state in lines 380-383: "Each axis of PCoA has an eigenvalue whose magnitude indicates the amount of variation in that axis. The proportion of a given eigenvalue to the sum of all eigenvalues reveals the relative importance of each axis. The confidence ellipse represents 95% of confidence level".

This is pretty similar to the explanation provided in the gustaME website (https://mb3is.megx.net/gustame/dissimilarity-based-methods/principal-coordinates-analysis). If that is the case, authors have to provide a reference for that statement.

3- Figure 4. Authors have corrected p-values using FDR approach and they have provided a new version for this figure (figure 4A). Nevertheless, the number of pathways found to differ significantly is higher than in the previous version in which no p-value correction was done. For example, endocrine system or environmental adaptation are significant after FDR correction but not without correcting for multiple comparisons. How is this possible? Have the authors removed any potential outlier in the new version of the figure?

Author Response

1- Authors have to provide a better explanation to the huge variability shown in sequencing depth, rather than it was performed following standard protocols. If this is a consequence of differences in bacterial load, this has to be confirmed by the authors. I strongly recommend the quantification of the DNA extracts using qPCR.

Response: We have confirmed this result from The Emory Integrated Genomic Core where the data were sequenced. The sequence counts of all samples ranged from 204,113 to 293,050 except one sample with sequence counts of 976,098, which should be associated with bacterial load. We checked about the data processing after dada2, our findings were not driven by this specific sample so we keep this sample in our study.

2- Figure 2. Homogeneity in the dispersion of the variance assumption in PERMANOVA results. Since part of the claims made by the authors are based on the results of this figure, it is critical to assess that the spread of the variance assumption is met using adequate methods. Authors stated in their response and the manuscript (lines 385-386): "The dispersion of the variance is homogenous in both PNS groups based on scatter plot and  independent sample t-test." T-test is not designed to check if two multivariate datasets are homoscedastic, but to check if there are differences between means coming from two normally distributed populations. Scatter plots are not statistical test either. If using R, authors can do this using the function betadisper provided in the package vegan. Authors have to provide the relevant values for the statistics as well as the p-values.

Also, in regards to one of my questions, authors state in lines 380-383: "Each axis of PCoA has an eigenvalue whose magnitude indicates the amount of variation in that axis. The proportion of a given eigenvalue to the sum of all eigenvalues reveals the relative importance of each axis. The confidence ellipse represents 95% of confidence level". This is pretty similar to the explanation provided in the gustaME website (https://mb3is.megx.net/gustame/dissimilarity-based-methods/principal-coordinates-analysis). If that is the case, authors have to provide a reference for that statement.

Response: The betadisper analysis shows the variance is homogenous in both PNS groups (F-score=0.343, p=0.564). We added the reference for PCoA as suggested, see references 72, 73.

3- Figure 4. Authors have corrected p-values using FDR approach and they have provided a new version for this figure (figure 4A). Nevertheless, the number of pathways found to differ significantly is higher than in the previous version in which no p-value correction was done. For example, endocrine system or environmental adaptation are significant after FDR correction but not without correcting for multiple comparisons. How is this possible? Have the authors removed any potential outlier in the new version of the figure?

Response: Thanks for pointing this out. We have double checked all the pathways to make sure they are all correct. For previous version, all pathways with significance smaller than 0.05 are labeled based on the raw p-value, some pathways are between 0.05 and 0.1 were not highlighted. After correction, pathway significance smaller 0.1 was labeled as well. All the labels are based on the corrected p-value as following: one star for <0.1, two stars for <0.05 and three stars for <0.01.

Round 4

Reviewer 3 Report

In the first version of the manuscript (no FDR correction), authors reported in Figure 4 11 differentially expressed pathways showing statistical significance (p<0.05 as per authors criterion for significance, line 419):

1- Xenobiotic biodegradation and metabolism *, p<0.05 (as per figure legend)

2- Transport and catabolism ***, p<0.001

3- Nervous system **, p<0.01

4- Membrane transport ***, p<0.001

5- Lipid metabolism *, p<0.05

6- Glycan biosyntehsis and metabolism ***, p<0.001

7- Folding, sorting and degradation **, p<0.01

8- Digestive system **, p<0.01

9- Cell motility **, p<0.01

10- Cancers: Overview *, p<0.05

11- Biosynthesis of other secondary metabolites **, p<0.01

In the new version of the manuscript, Figure 4A shows the same 11 features with a p<0.05 after correction for multiple comparisons using FDR.

1- Xenobiotic biodegradation and metabolism **=p<0.05 (as per figure legend)

2- Transport and catabolism ***, p<0.01

3- Nervous system **, p<0.05

4- Membrane transport **, p<0.05

5- Lipid metabolism *, p<0.05

6- Glycan biosyntehsis and metabolism ***, p<0.01

7- Folding, sorting and degradation **, p<0.05

8- Digestive system **, p<0.05

9- Cell motility **, p<0.05

10- Cancers: Overview **, p<0.05

11- Biosynthesis of other secondary metabolites **, p<0.05

It is impossible that the same number features are significant before and after FDR correction, as well as how those p-values change. Please, provide a table with: mean, standard deviation and number of samples per group, the original p-values for every single comparison, and the FDR-corrected p-value.

Author Response

Reviewer 3

In the first version of the manuscript (no FDR correction), authors reported in Figure 4 11 differentially expressed pathways showing statistical significance (p<0.05 as per authors criterion for significance, line 419):

1- Xenobiotic biodegradation and metabolism *, p<0.05 (as per figure legend)

2- Transport and catabolism ***, p<0.001

3- Nervous system **, p<0.01

4- Membrane transport ***, p<0.001

5- Lipid metabolism *, p<0.05

6- Glycan biosyntehsis and metabolism ***, p<0.001

7- Folding, sorting and degradation **, p<0.01

8- Digestive system **, p<0.01

9- Cell motility **, p<0.01

10- Cancers: Overview *, p<0.05

11- Biosynthesis of other secondary metabolites **, p<0.01

In the new version of the manuscript, Figure 4A shows the same 11 features with a p<0.05 after correction for multiple comparisons using FDR.

1- Xenobiotic biodegradation and metabolism **=p<0.05 (as per figure legend)

2- Transport and catabolism ***, p<0.01

3- Nervous system **, p<0.05

4- Membrane transport **, p<0.05

5- Lipid metabolism *, p<0.05

6- Glycan biosyntehsis and metabolism ***, p<0.01

7- Folding, sorting and degradation **, p<0.05

8- Digestive system **, p<0.05

9- Cell motility **, p<0.05

10- Cancers: Overview **, p<0.05

11- Biosynthesis of other secondary metabolites **, p<0.05

It is impossible that the same number features are significant before and after FDR correction, as well as how those p-values change. Please, provide a table with: mean, standard deviation and number of samples per group, the original p-values for every single comparison, and the FDR-corrected p-value.

Response: Thanks for this comment. We have revised the pathways and added a Table S1 based on this comment. Thanks.

Round 5

Reviewer 3 Report

Based on the Table S1, it looks like that the p-values in the previous version of figure 4 were wrong.

This now makes sense. I do not have any other question for the authors.